# Smurf1 Suppression Enhances Temozolomide Chemosensitivity in Glioblastoma by Facilitating PTEN Nuclear Translocation

**DOI:** 10.3390/cells11203302

**Published:** 2022-10-20

**Authors:** Lei Dong, Yang Li, Liqun Liu, Xinyi Meng, Shengzhen Li, Da Han, Zhenyu Xiao, Qin Xia

**Affiliations:** School of Life Science, Beijing Institute of Technology, Beijing 100081, China; ldong@bit.edu.cn (L.D.); cherylliyang@126.com (Y.L.); 2793221771@qq.com (L.L.); 1004382302@qq.com (X.M.); lsz1846176947@163.com (S.L.); handa1994@163.com (D.H.); xiaozy@bit.edu.cn (Z.X.)

**Keywords:** TMZ, drug resistance, glioblastoma, smurf1, PTEN

## Abstract

The tumor suppressor PTEN mainly inhibits the PI3K/Akt pathway in the cytoplasm and maintains DNA stability in the nucleus. The status of PTEN remains therapeutic effectiveness for chemoresistance of the DNA alkylating agent temozolomide (TMZ) in glioblastoma (GB). However, the underlying mechanisms of PTEN’s interconnected role in the cytoplasm and nucleus in TMZ resistance are still unclear. In this study, we report that TMZ-induced PTEN nuclear import depends on PTEN ubiquitylation modification by Smurf1. The Smurf1 suppression decreases the TMZ-induced PTEN nuclear translocation and enhances the DNA damage. In addition, Smurf1 degrades cytoplasmic PTEN K289E (the nuclear-import-deficient PTEN mutant) to activate the PI3K/Akt pathway under TMZ treatment. Altogether, Smurf1 interconnectedly promotes PTEN nuclear function (DNA repair) and cytoplasmic function (activation of PI3K/Akt pathway) to resist TMZ. These results provide a proof-of-concept demonstration for a potential strategy to overcome the TMZ resistance in PTEN wild-type GB patients by targeting Smurf1.

## 1. Introduction

Glioblastoma (GB) is one of the most common malignant brain tumors, accounting for about 45% of primary malignant brain tumors and 15% of central nervous system tumors [1,2]. Despite extensive research efforts to better understand and treat these tumors, the prognosis for GB patients treated with Stupp standard procedures (surgery followed by radiation and chemotherapy) remains poor, with a median survival time fewer than 15 months [3]. Temozolomide (TMZ), which is more toxic in cancer cells with a slightly more alkaline pH than in normal cells, is the first-line chemotherapy agent in the treatment of GB [4]. However, the development of TMZ resistance often becomes the limiting factor in effective treatment. Thus, investigating mechanisms of TMZ resistance can help to identify novel drug targets and provide effective chemotherapies. Previous studies show that TMZ promotes the methylation at N^7^-guanine (the most common, 70%), followed by O^6^-guanine (6%, critical for tumor cytotoxic activity), and O^3^-Adenine (9%) [5]. The mismatch repair (MMR), which recognizes and removes the nucleotide mismatch (O^6^ methylguanine thymine) in the newly synthesized DNA chain, plays a vital role in the sensitivity of TMZ by the formation of accumulated DNA double-strand breaks, ultimately leading to cell apoptosis. Thus, the MMR deficiency causes TMZ resistance by mediating the formation of O^6^-methylated guanine (MG)-containing mismatches to facilitate the adaptive gene mutation [6]. There are other reasons accounting for TMZ resistance: (1) The ABC transporter from the blood–brain barrier (BBB) and glioma stem cells (GSCs) export foreign TMZ out of cells, leading to limited capacity to access the BBB (20% of its levels in the systemic circulation) [7,8]. (2) The high level of methylation of O6-methylguanine-DNA methyltransferase (MGMT) expression directly accounts for resistance by repairing the TMZ-induced O^6^-MG. The patients (approximately 60% of GB) with unmethylated MGMT have a lower survival rate [9,10]. The MGMT inhibitor O^6^-benzyl guanine upregulates the GB sensitivity to TMZ [5,11,12]. (3) Tumor DNA repair systems, such as the base excision repair (BER) and DNA strand break (DSB) repair, account for TMZ-induced alkylation and DSB failure [6,13,14,15,16]. In addition, within BER modulation, poly ADP ribose polymerase-1 (PARP-1) is essential for the recruitment of BER proteins and consequent DNA repair [13]. PARP-1 also promotes the activation of DSB repair proteins such as phosphorylated histone H2A.X, p53, and SMC1 by interaction with the DNA damage response kinase Ataxia telangiectasia mutated (ATM) [17]. The PARP-1 inhibitor overcomes TMZ resistance in MMR-deficient primary GB cells that are independent of BER [6]. A further study verifies that PARP-1 can also mediate PARylation of MGMT to TMZ resistance by repairing O^6^-MG DNA damage in GB [9].

The status of PTEN in GB (35% mutation rate) greatly influences TMZ resistance. By combining the EGFR tyrosine kinase inhibitor erlotinib with radiation therapy and TMZ, GB patients with MGMT gene silencing and intact PTEN have a significant survival advantage [18]. Studies show that the TMZ treatment is more effective in eradicating GB with PTEN loss [19], suggesting that PTEN is another molecular signature to affect GB patient survival. Mechanistically, cytoplasmic PTEN exerts its lipid phosphatase activity and dephosphorylates PIP3 to PIP2 in order to block PI3K pathways [20,21,22]. For example, the activation of the PI3K/Akt pathway increases the expression of PTEN, and the perturbations of PTEN status (mutation) limit the efficiency of PI3K inhibitors (BYL-719 and AZD8186) in tumors [23]. The combination of PI3Ki/PARPi presents an efficient therapeutic approach in PTEN-deficient tumors [24]. In addition, accumulating evidence has shown that nuclear PTEN is involved in the regulation of DNA damage repair, chromosome stability, and cell-cycle progression in its phosphatase-independent manner. The loss of PTEN leads to the accumulation of DSBs and genomic instability by impairing CHK1 function [25]. Nuclear PTEN promotes the tumor-suppressive activity of the APC-CDH1 complex for the fail-safe cellular senescence response [26]. 

The modification of PTEN is critical for cytoplasmic and nuclear localization and functions as a tumor suppressor. The existing studies show that post-translational modifications, such as mono-ubiquitination of PTEN by Nedd4 and de-ubiquitination by HAUSP/USP7, contribute to the nuclear import and export of PTEN, respectively [27,28,29]. Monoubiquitylation (K13 or K289), phosphorylation (S113), and SUMOylation (K254) promote PTEN nuclear translocation [30,31,32,33]. Zhang et al., mentioned that neddylation (K197 and K402) of PTEN regulates its nuclear import and promotes tumor development by dephosphorylation and stabilization of the fatty acid synthase [32]. It has been observed that the blockage of PTEN nuclear import promotes glioma sensitivity to chemo- or radiotherapy. For instance, the inhibition of nuclear PTEN by blocking phosphorylation of Y240-PTEN enhances the sensitivity to radiotherapy efficacy through attenuated DNA repair [34,35]. It also suggests the potential regulatory mechanisms that specifically regulate the abundance of nuclear PTEN in response to TMZ treatment. However, these mechanisms have rarely been studied. Elucidating this mechanism will help to learn the relationship between nuclear PTEN and TMZ drug resistance and provide a new perspective on the therapeutic depletion of nuclear PTEN in combination with TMZ.

The HECT-type E3 ubiquitin ligase Smad ubiquitylation regulatory factor 1 (Smurf1) was initially identified to degrade SMADs through ubiquitination and the 26S proteasome in the TGF-β/BMP pathway [36]. Existing studies reported an inhibitory role of Smurf1 in cancer metastasis in lung cancer by targeting SRSF5 [37]. Other studies demonstrated that Smurf1 regulates multiple substrates, including UVRAG, Kindlin-2, ER-alpha, and p120-catenin, to promote tumor progression in different cancers [38,39,40,41]. Elevated Smurf1 in GB correlates with a worse prognosis [42]. We previously reported that Smurf1 suppression decreases the ubiquitination and degradation of PTEN to inhibit the PI3K/Akt pathway [43]. However, whether Smurf1 regulates PTEN nuclear import and drug resistance remains unclear. The obtained results show that TMZ promotes the nuclear import of PTEN dependent on Smurf1. Smurf1 knocking down decreases PTEN nuclear translocation, leading to enhanced TMZ-induced DNA damage. In summary, this study explores the therapeutic implications of Smurf1 and TMZ resistance, which may provide a promising treatment for GB.

## 2. Materials and Methods

### 2.1. Cell Culture and Transfections

GB cell lines LN229 (CRL-2611) and U87 (HTB-14) were purchased from American Type Culture Collection. GB cell lines U251 (HTX1725) and U343 (HTX2007) were purchased from Otwo Biotech (Shenzhen, China). Cells were transfected with lentiviral vector-harboring shScramble (5′-CCTAAGGTTAAGTCGCCCTCGCTCGAGCGAGGGCGACTTAACCTTAGG-3′) and Smurf1-shRNA (Open Biosystems) (5′-GCCCAGAGATACGAAAGAGAT-3′) using the VigoFect transfection reagent (Vigorous Biotechnology Beijing Co., Ltd.) according to the manufacturer’s instructions. After obtaining the lentivirus as described above, stable knockdown of endogenous human Smurf1 was achieved by lentivirus infection. All cells were cultured in DMEM (C11995500BT, Gibco) with fetal bovine serum (10%) and antibiotics penicillin (100 U/mL)/streptomycin (100 μg/mL) in an incubator (5% CO_2_, 37 °C). 

The TMZ-resistant glioma cell line (LN229R) was obtained by gradually exposing the parental cell line (LN229) to medium with increasing doses of TMZ (2–100 µM, 85622-93-1, MACKLIN) over 2 months. The medium containing TMZ was changed every 2–3 days. In addition, 250 μM TMZ was used for subsequent analysis. 

Primary MEFs were obtained from embryos at 14–16 days post coitum. The uterine horns of anesthetized pregnant mice were rinsed in ice-cold sterile PBS. The embryos were separated from their placenta and placed in a 60 mm plate containing ice-cold sterile PBS. The head, liver, and gut from the embryo were removed and the remaining portion put in a 60 mm plate containing trypsin EDTA. Then, they were cut into small pieces (~1 mm^3^) using a sterile razor blade and scissors. The chopped materials were incubated with 5 mL of trypsin EDTA for 30 min at 37 °C. Then, 1 mL DMEM (with FBS and penicillin/streptomycin) was added to the chopped embryos, followed by centrifuging at 1000 rpm speed for 5 min. The pellets were resuspended and transferred to a 60 mm dish containing DMEM (with FBS and penicillin/streptomycin). 

Cells were cultured in 12-well plates and grown for 12 h before treatments with TMZ. Cell numbers were recorded by TC10 Automated Cell Counter (Bio-Rad, Hercules, California, USA) at indicated days.

### 2.2. MTT Assay

The cells were seeded at a density of 10^4^ cells/well in 96-well plates. The cells were incubated at 37 °C, then 10 μL MTT (10 mg/mL, Solarbio, Beijing, China, M8180) was added to wells for 4 h at 37 °C. Finally, the 150 μL DMSO was added, and the absorbance was measured at the wavelength of 490 nm on the microplate reader to analyze cell proliferation and viability.

### 2.3. Plasmids, si-RNA, and Transfections

pCMV-PTEN was obtained from Addgene. A full-length PTEN cDNA was amplified from pCMV-PTEN, then was loaded onto 3×Flag vector. The point mutation of Flag-PTEN was generated by the site-directed mutagenesis with the following primers: 5′-TCAGAAGAAGTAGAAAATGGA-3′ and 5′-GGTTTCCTCTGGTCCTGGTAT-3′ for Flag-PTEN^K289E^. The mutations were confirmed by sequence analysis. 

The human siRNAs were purchased from JTSBIO (Wuhan, China): si-Smurf1 was 5′-GCGUUUGGAUCUAUGCAAATT-3′, si-Smurf1-1 was 5′-CCAGGGAGUGGCUUUACUUTT-3′, si-PTEN was 5′-CCACCACAGCUAGAACUUATT-3′, and si-PTEN-2 was 5′-GGTGTAATGATATGTGCAT-3′. 

The cells were transfected with plasmids or siRNAs using Lipofectamine 2000 (Invitrogen, Waltham, MA, USA) or RNAiMAX reagent (Invitrogen, Waltham, MA, USA) following the supplier’s instructions, respectively. The overexpression or knockdown efficiency was examined by Western blotting. 

### 2.4. Mice

Mice were housed in specific pathogen-free facilities, and the Ethics Review Committee for Animal Experimentation of the Beijing Institute of Technology University approved the experimental protocol. Smurf1^WT^ and Smurf1^KO^ mice were a kind gift from Dr. Lingqiang Zhang (Beijing Institute of Lifeomics, China). 

The cells were inoculated subcutaneously into Male BALB/c nude mice (6–7 weeks old; SPF (Beijing) Biotechnology Co., Ltd.). The length (L) and width (W) of the tumor were measured with a caliper every 7 days, and tumor volumes were calculated using the equation volume = (π × L × W^2^)/6. For TMZ treatment, mice were intraperitoneally injected with 20 mg/kg TMZ every 2 days for 5 weeks. The mice were anesthetized and euthanized, and the tumors were removed, imaged, and weighed. The Beijing Institute of Technology University Institutional Animal Care and Use Committee approved all animal studies in this study. The study is compliant with all relevant ethical regulations involving the manipulation of experimental animals.

### 2.5. Western Blotting

The RIPA buffer (50 mM pH 7.6 Tris–HCl, 150 mM NaCl, 0.5% sodium deoxycholate, 1% NP-40, and protease inhibitor cocktail from Roche) with PMSF and phosphatase inhibitor was used to lyse cells. The samples were separated via SDS-PAGE and transferred to nitrocellulose filter membrane. The nitrocellulose filter membrane was blocked by 5% skim milk, followed by incubating with primary antibodies (12 h) and peroxidase-conjugated secondary antibodies (1 h). Finally, the bands were visualized by chemiluminescence reagents.

Primary antibodies: β-actin (1:2000, Sigma, Taufkirchen, Germany, A1978-200), Smurf1 (1:500, Santa Cruz, Dallas, TX, USA, sc100616), p-p70 S6 kinase (Thr389) (1:500, Cell Signaling Technology, Danvers, MA, USA, #9205), p70 S6 kinase (1:500, Santa Cruz, Dallas, TX, USA, sc-8418), p-Akt (Ser473) (1:1000, Cell Signaling Technology, Danvers, MA, USA, #4060), Akt (1:1000, Cell Signaling Technology, Danvers, MA, USA, #9272), p-mTOR (Ser2448) (1:500, Cell Signaling Technology, Danvers, MA, USA, #2971), mTOR (1:1000, Cell Signaling Technology, Danvers, MA, USA, #2983), PTEN (1:500, Santa Cruz, Dallas, TX, USA, sc-7974), Flag (1:1000, Sigma, Taufkirchen, Germany, F3165), PARP (1:500, Santa Cruz, Dallas, TX, USA, sc-8007), Ub (1:1000, Abclonal, Wuhan, China, A19686), and γH2A.X (Ser139) (1:1000, Abcam, Cambridge, UK, ab26350). Secondary antibodies were used at 1:5000 dilution: Horseradish Peroxidase conjugated goat anti-rabbit IgG (BOSTER, Pleasanton, CA, USA, BA1054), Horseradish Peroxidase conjugated goat anti-mouse IgG (BOSTER, Pleasanton, CA, USA, BA1050). Relative protein levels were quantified by scanning densitometry, and the relative gray value of proteins corrected for background was calculated as: (band intensity of protein of interest)/(band intensity of loading control).

### 2.6. Immunofluorescence

The cells were cultured on coverslips and fixed with 4% paraformaldehyde for 15 min and washed twice with PBS. The slides were treated with 0.1% TritonX-100 (PBS) for 5 min and washed twice with PBS, then blocked with 5% BSA for 1 h. Subsequently, the slides were incubated by primary antibody Flag (1:500, Sigma, Taufkirchen, Germany, F3165), PTEN (1:200, Santa Cruz, Dallas, TX, USA, sc-7974), or γH2A.X (Ser139) (1:1000, Abcam, Cambridge, UK, ab11174) for 12 h and fluorescently labeled by secondary antibody Alexa Fluor^®^ 488 goat anti-mouse IgG (Life Technologies, Waltham, MA, USA, A11001) for 1 h. The cell nuclear was stained by Fluor shield mounting medium with DAPI (Abcam, Cambridge, UK). The samples were observed on Nikon N-SIM microscope. 

### 2.7. Cytoplasmic and Nuclear Protein Extraction

Cells were collected and lysed with sucrose buffer with 0.5% NP40 for 40 min on ice (sucrose buffer: 0.32 M sucrose, 3 mM CaCl_2_, 2 mM MgAc, 0.1 M EDTA, 1 mM DTT, 0.5 mM PMSF). The lysate was separated via centrifuging at 600× *g* for 15 min at 4 °C. The supernatant was collected as cytoplasm. The precipitate was resuspended with sucrose buffer and centrifuged at 600× *g* for 15 min at 4 °C 4 times. The precipitate was resuspended with RIPA buffer with PMSF and phosphatase inhibitors as the nucleus.

### 2.8. Immunoprecipitation

The sample was centrifuged at 12,000 rpm for 10 min at 4 °C. The rProtein G beads (Solarbio, Beijing, China, R8300) were washed with PBS at 13,500× *g* for 2 min at 4 °C 3 times. The anti-PTEN (Santa Cruz, Dallas, TX, USA, sc-7974) was added to beads and incubated for 4 h at 4 °C. Then, the antibody was removed, and the beads were washed with PBS at 3000× *g* for 2 min at 4 °C 6 times. The sample was incubated with the beads for 12 h at 4 °C. After the incubation, the beads were washed with 1 × PBS 4 times. Both samples were identified by Western blotting.

## 3. Results

### 3.1. Combination of Smurf1 Suppression with TMZ Synthetically Inhibits PI3K/Akt Pathway

We previously reported PI3K/Akt signaling is prohibited in LN229 (PTEN wild type), compared to U343 (PTEN wild type) and U87/U251 (PTEN deficiency) [42]. A previous report showed that the upregulated PI3K/Akt pathway contributes to TMZ resistance [44]. It reported that RNA modification regulator adenosine deaminases acting on RNA (ADARs) promote TMZ resistance by upregulating p-Akt. U343 expressing low ADARB1 may restore TMZ sensitivity in GB cells by decreased p-Akt [45]. To study whether the status of PTEN and/or Akt signaling affects the sensitivity of TMZ, we first analyzed the publicly available dataset in the Genomics of Drug Sensitivity in Cancer (GDSC) and The Cancer Genome Atlas Program (TCGA-GB) where GB samples were classified into two groups according to status of PTEN: wild type (*n* = 16 in GDSC-GB and *n* = 102 in TCGA-GB) and mutant (*n* = 17 in GDSC-GB and *n* = 47 in TCGA-GB). Results support the idea that there is no statistical difference in TMZ IC50 between the PTEN WT and PTEN mutant group (Figure 1a,b). Specifically, the PTEN^Low^ subgroup (*n* = 12 in TCGA-GB) had significantly worse survival compared with the PTEN^High^ subgroup (*n* = 17 in TCGA-GB) in PTEN wild-type patients with TMZ treatment (Figure 1c). Thus, we propose that PTEN wild-type expression made a difference on TMZ resistance. Additionally, the IC50 for TMZ of the U343 cell line was not included in the GDSC dataset. We then tested the cytotoxic effect of TMZ in the LN229, U251, U343, and U87 cell lines. The IC_50_ for TMZ of cell lines was as follows: LN229, 450 ± 50 μM; U87, 285 ± 40 μM; U251, 269 ± 40 μM; U343, 31 ± 10 μM (Figure A1a,b). These data showed that LN229, but not U343, was more resistant to TMZ treatment than the PTEN-deficient U87 and U251 cell lines. We also treated LN229, U343, U87, and U251 cell lines with TMZ, finding that LN229, but not U343, was more resistant to TMZ treatment than the PTEN-deficient U87 and U251 cell lines (Figure 1d). We then assumed that TMZ feedback induces the activation of PI3K/Akt signaling. Indeed, we found that TMZ hyperactivated the PI3K/Akt pathway in LN229, U251, and U87 cells, evidenced by upregulation of p-Akt^S473^, p-mTOR^S2448^, and p-p70S6K^T389^, but not U343 (Figure 1e). These results suggest the inability to activate the PI3K/Akt pathway may be associated with U343 sensitivity to TMZ.

Smurf1 is overexpressed in GB cells and promotes cell growth. Given that our previous report showed that Smurf1 suppression decreases the ubiquitination and degradation of PTEN to inhibit the PI3K/Akt pathway [43], we analyzed the overall survival in the Smurf1 high group (*n* = 29, including 24 patients (*n* = 23 with wild-type PTEN) with TMZ treatment) and Smurf1 low group (*n* = 30, including 28 patients (*n* = 20 with wild-type PTEN) with TMZ treatment) in the TCGA GB patient dataset, and found that the Smurf1 high group had significantly worse survival compared with low group (Figure 1f). To evaluate whether suppression of Smurf1 could restore cell growth and sensitize the cells to TMZ treatment, we transfected GB cells with si-Smurf1 and si-Control RNA and treated with or without TMZ. Consistently, we found no statistical difference in the efficacy of treatment of PTEN-deficient U251 and U87 cells between the dual-treatment group (si-Smurf1 + TMZ) and the TMZ alone group (Figure 1g). Importantly, the dual-treatment group showed reduced cell survival compared to the TMZ alone group in both LN229 and U343 cells (Figure 1g). Further, PTEN knockdown reduced and PTEN overexpression restored the effect of dual treatment in LN229 and U251, respectively, implying that Smurf1 promotes TMZ resistance in a PTEN-dependent manner (Figure A1c,d).

We then knocked down Smurf1 in LN229 and U251 to investigate whether suppression of Smurf1 sensitizes TMZ treatment by inhibiting the PI3K/Akt pathway. We found that the activation of the PI3K/Akt pathway is inhibited in the dual-treatment compared with the TMZ alone treatment in LN229, but not in U251 (Figure 1h,i). Of note, PTEN knockdown inhibited si-Smurf1-reduced suppression of the PI3K/Akt pathway in LN229 (Figure A1e–g), suggesting that suppression of Smurf1 inhibits Akt signaling in a PTEN-dependent manner. We next investigated whether Smurf1-mediated hyperactivation of PI3K/Akt depends on nuclear PTEN. We transfected Flag-PTEN^K289E^, which is defective in nuclear import [46], into U251 cells and found that Smurf1 knockdown still decreased p-Akt^S473^, p-mTOR^S2448^, and p-p70S6K^T389^, demonstrating that Smurf1 degrades cytoplasmic PTEN to activate PI3K/Akt under TMZ treatment (Figure 1j), and the inhibition of Smurf1 to prohibit Akt signaling is nuclear PTEN independent. More importantly, Smurf1 knockdown combined with TMZ treatment significantly decreased the cell proliferation of U251 with Flag-PTEN^K289E^ (Figure 1k). These results illustrate that Smurf1-mediated hyperactivation of PI3K/Akt is sufficient to induce TMZ resistance independent of nuclear PTEN.

### 3.2. Smurf1 Facilitates PTEN Nuclear Translocation to Repair DNA Damage under TMZ Treatment

PARP-induced phosphorylated H2A.X (γH2A.X) is one of the DNA damage markers by instant accumulation at nascent DSB sites [13,47]. Interestingly, we found the dual treatment led to an increase in PARP and γH2A.X in LN229, while the TMZ alone group and treated U251 cells showed no similar increase (Figure 2a–d and Figure A2a,b), suggesting suppression of Smurf1 may also promote DNA damage. Of note, transfection of PTEN in U251 had no effect on PARP and γH2A.X expression between the si-Smurf1 and si-Control group (Figure A2c,d), suggesting that TMZ is the inducer of the DNA damage. To further confirm the protective role of nuclear PTEN in TMZ-induced DNA damage, we transfected Flag-PTEN or Flag in U251, and si-PTEN or si-Control RNA in LN229, respectively. By overexpression of PTEN in U251, Smurf1 suppression rescued the promotion effect by increased γH2A.X and PARP protein levels in response to TMZ (Figure 2e,f). Conversely, the combination of si-Smurf1 and TMZ failed to induce the synthetic effect in LN229 with knocking down of PTEN (Figure 2g,h and Figure A2e–h). These results imply that Smurf1-suppression-enhanced TMZ-induced DNA damage is due to the nuclear translocation of PTEN.

To investigate whether Smurf1 increased the nuclear translocation of PTEN under TMZ treatment, we performed a cytoplasmic and nuclear protein extraction experiment in LN229 and U251-Flag-PTEN cells. TMZ treatment increased the nuclear fraction of PTEN and decreased the cytoplasmic proportion of PTEN in LN229 under TMZ treatment (Figure 2i–l), suggesting that TMZ induces PTEN nuclear translocation. Notably, Smurf1 knockdown significantly inhibited PTEN nuclear translocation in LN229 and U251-Flag-PTEN in TMZ treatment (Figure 2i–l), indicating that TMZ-induced PTEN nuclear translocation is dependent on Smurf1. To quantify the percentage of PTEN^+^ cells with or without TMZ treatment in the si-Smurf1 or si-Control groups, we separated PTEN^+^ cells into three groups: (1) nuclear PTEN^+^, cytoplasmic PTEN is less than 35% nuclear PTEN immunofluorescent intensity (cytoplasm < nuclear), (2) 65% nuclear > cytoplasm > 35% nuclear, (3) cytoplasmic PTEN^+^, cytoplasm > 65% nuclear. We found that the percentage of nuclear PTEN^+^ cells (around 60%) was significantly decreased by suppression of Smurf1 under TMZ treatment in U251-Flag-PTEN, U343, and LN229 cells (Figure A2i–m), which further demonstrates that Smurf1 promotes nuclear PTEN import. Next, we studied whether PTEN nuclear import depends on PTEN ubiquitylation modification by Smurf1. LN229 cells were treated with TMZ, and immunoprecipitation was performed using a PTEN antibody. The nuclear fraction was subjected to SDS-PAGE and probed with the Ub antibody. TMZ treatment significantly increased the Ub level of nuclear PTEN (Figure 2m). Furthermore, Smurf1 knockdown significantly reduced the ubiquitination level of nuclear PTEN in LN229 with TMZ treatment (Figure 2m), suggesting that Smurf1 mediates PTEN modification by its ubiquitylation. These data indicate that Smurf1 suppression downregulates TMZ-induced nuclear import of PTEN by inhibiting its ubiquitination. 

It has been shown that the enhanced nuclear PTEN accumulation is associated with increased mono-ubiquitination at the K289 site [46]. To investigate whether Smurf1 promotes TMZ-induced PTEN nuclear import by K289 mono-ubiquitylation, U251 cells were transfected with Flag-PTEN^K289E^ followed by TMZ treatment. Consistently, TMZ promotes PTEN wild-type nuclear import, but not PTEN^K289E^, suggesting that TMZ promoted PTEN nuclear import by mono-ubiquitination at the K289 site (Figure A2n). The cytoplasmic and nuclear protein extraction of PTEN^K289E^ expressing U251 with TMZ treatment showed that Smurf1 knockdown has no significant difference in the protein level of γH2A.X, indicating that ubiquitination of K289 PTEN plays an essential role in Smurf1-mediated DNA repair (Figure 2n,o). These results implied that Smurf1 facilitates PTEN nuclear translocation to repair DNA damage under TMZ treatment by K289 mono-ubiquitination.

### 3.3. Smurf1 Suppression Restores the Sensitivity of TMZ-Resistant GB Cells

To investigate the efficacy of Smurf1 suppression in TMZ-resistant GB cell lines, we cultured LN229R and U343R cells, which are TMZ resistant, by exposing LN229 or U343 cells to stepwise increasing concentrations of TMZ (2–100 µM) over 2 months. The cell viability assay showed that LN229R or U343R had significantly higher TMZ tolerance than LN229 or U343 cells (Figure 3a and Figure A3a). Next, we knocked down Smurf1 in LN229 and LN229R under TMZ treatment. The results showed that the dual-treatment group decreased cell survival compared to the single TMZ treatment in LN229R, implying that Smurf1 suppression inhibited the proliferation of TMZ resistance cells (Figure 3b). Additionally, LN229R showed an increased significant PI3K/Akt pathway activation compared with parental LN229 cell lines, evidenced by upregulation of p-Akt^S473^, p-mTOR^S2448^, and p-p70S6K^T389^ (Figure 3c). To test the underlying effectiveness of si-Smurf1 in TMZ re-sensitivity by enhanced DNA damage and suppression of Akt signaling, we transfected LN229R and U343R with si-Smurf1. We found that dual treatment of siSmurf1 and TMZ increased γH2A.X and reduced PI3K/Akt (p-Akt^S473^, p-mTOR^S2448^, and p-p70S6K^T389^) compared with the TMZ treatment alone (Figure 3d–f and Figure A3b–e). Importantly, it also showed that PTEN knockdown diminished the effectiveness of si-Smurf1 in LN229R and U343R, suggesting LN229R and U343R were re-sensitive to TMZ by targeting Smurf1 in a PTEN-dependent manner (Figure 3d–g and Figure A3f–i). Taken together, Smurf1 interconnectedly regulates PTEN both to activate PI3K/Akt (degradation of cytoplasmic PTEN) and repair DNA damage (import of nuclear PTEN) responding to TMZ treatment. 

### 3.4. Smurf1 Knockout Restores TMZ Sensitivity of PTEN Wild-Type GB Cells In Vivo

To examine the combined effect of Smurf1 suppression and TMZ in vivo, mouse embryonic fibroblasts (MEFs) were generated from Smurf1^+/+^ mice and Smurf1^−/−^ mice. Consistently, TMZ-treated Smurf1^−/−^ MEFs displayed a decreased PI3K/Akt pathway (downregulation of p-Akt^S473^, p-mTOR^S2448^, p-p70S6K^T389^) and induced DNA damage (upregulation of PARP and γH2A.X) compared to TMZ-treated WT MEFs (Figure 4a–c). To further explore the synthetically lethal effect, the LN229-shSmurf1 and LN229-shScramble cells were subcutaneously injected into the left and right flanks of eight-week-old female nude mice. Fourteen days after implantation, the mice were treated intraperitoneally with TMZ or normal saline every two days for five weeks (Figure 4d, e). Significantly, TMZ suppressed the growth rate, size, and weight of tumors in LN229-shSmurf1 compared to in LN229-shScramble, suggesting that the combination of Smurf1 inhibition and TMZ treatment synthetically inhibited tumor growth (Figure 4f–h). In sum, Smurf1 interconnectedly regulates PTEN both to activate PI3K/Akt (degradation of cytoplasmic PTEN) and repair DNA damage (import of nuclear PTEN) to resist TMZ. The co-treatment of Smurf1 suppression and TMZ has potential efficacy for PTEN wild-type GB (Figure 5).

## 4. Discussion

Clinical findings suggest that PTEN gene alterations are associated with poor prognosis and may influence tumor responses to current therapies in high-grade GB [48]. Previous reports show only 25% of cancer patients showing a correlation between loss of PTEN and loss of its mRNA [49,50]. Our result indicates that understanding the protein levels and localization of PTEN in patient tumors may further help stratify these patient populations. According to a relevant survey, 7.1% of GB patients manifested nuclear positivity for PTEN [51]. Several studies reported that nuclear PTEN plays a crucial role in regulating DNA damage repair [34]. However, the role of nuclear PTEN in response to chemotherapy resistance remains largely unknown. Recent studies demonstrated that ionizing radiation (IR) induces drug resistance by promoting nuclear PTEN import independent of its lipid phosphatase activity [34]. This study revealed that Smurf1 contributes to TMZ chemoresistance by promoting PTEN nuclear import. PTEN nuclear translocation is a dynamic process [52]. Ubiquitination, SUMOylation, and phosphorylation have been shown to contribute to the translocation of PTEN. A previous study has proven that E3 ligase Nedd4-1, WWP1, WWP2, and Smurf1 target cytoplasmic PTEN for ubiquitin-mediated proteasome degradation [43,53,54,55]. Monoubiquitylation of PTEN by Nedd4-1 also contributes to PTEN nuclear import [46,52]. Therefore, this study aims at studying whether Smurf1 selectively induced ubiquitylation of PTEN to promote PTEN nuclear import. Here, we showed that Smurf1 knockdown reduces nuclear PTEN import in U251 with PTEN overexpression. In addition, Smurf1 knockdown decreases the protein level and ubiquitylation of nuclear PTEN in response to TMZ in PTEN wild-type LN229, demonstrating that Smurf1 promotes ubiquitylation of PTEN and its nuclear import. Furthermore, we identified that TMZ promotes PTEN nuclear import by mono-ubiquitination at the K289 site, which may be mediated by Smurf1.

Therefore, what is the role of the imported nuclear PTEN in response to TMZ? Previous studies have shown that nuclear PTEN is involved in maintaining genome stability by promoting DNA repair. They also demonstrated that the modifications of PTEN in the nucleus play a controversial role in tumor development. The phosphorylation of PTEN at Y240 sites promotes the recruitment of RAD51 to accelerate DNA repair by facilitating its interaction with Ki-67 [34]. A SUMOylation of PTEN at K254 is required for homologous recombination repair of DSBs through the DNA-damage-induced protein kinase ATM cascade [33]. However, the neddylation of PTEN at K197 and K402 sites by XIAP promotes tumorigenesis and progression through its interaction with FASN, in order to increase de novo fatty acid synthesis [32]. In this case, we proposed that the Smurf1-induced PTEN nuclear translocation to facilitate resistance for TMZ may also be mediated by chromosomal stability and DNA repair capacity. Indeed, Smurf1-mediated PTEN nuclear translocation protects DNA against being damaged, which is indicated by the decreased DNA DSBs marker γH2A.X. Notably, PARP-1/2 inhibitor ABT-888 enhances the TMZ efficacy in PTEN-deficient GB [48]. Consistently, similar to PTEN-deficient lines (U251 and U87), the PTEN wild type line (LN229) with Smurf1 knockdown also has high protein level of DNA damage signaling protein PAPR-1 and γH2A.X in response to TMZ.

Accumulating evidence shows that the PI3K/Akt/mTOR pathway is activated in response to TMZ, which is another essential factor in drug resistance [56,57]. In addition to their critical roles in balancing anabolic and catabolic responses, Akt and its downstream effector mTORC1 react to changes in genomic integrity [58,59]. Akt has been implicated in regulating diverse substrates that have nuclear functions, including important roles in DNA damage response (DDR), DNA repair, and the maintenance of genomic stability [60]. The study demonstrates that DSB promotes accumulation and co-localization of p-Akt^S473^ with γH2AX and p-ATM^S1981^, which makes non-homologous end-joining-mediated DSB repair start [61,62]. Activated Akt in the nucleus (immediately after Ser473 phosphorylation by DNAPK) conceivably regulates phosphorylation of the transcription factor FOXO, which resides in the nucleus prior to Akt phosphorylation [63,64], or DNA repair effector proteins, such as XLF or MERIT40 [59,65]. Moreover, Akt shuttles between nuclear and cytoplasmic compartments and thereby carries a nuclear signal (via phosphorylation) back out to the cytoplasm (inside-out signaling) to phosphorylate its downstream targets. Inhibition of mTORC1 signaling enhances DNA damage sensitivity [66]. Thus, DNA-damage-induced Akt activation expedites crosstalk between DNA repair/genome stability and cell growth/survival pathways [67]. In addition, inhibiting cellular DNA repair factors block Akt activation and promote reactivation. Whether PTEN affects Akt activity by repairing DNA damage deserves further confirmation. The interaction between nuclear PTEN (as DNA repair factor) and Akt requires further study. Similar to Nedd4-1 and WWP1, Smurf1 can ubiquitylate and degrade cytoplasmic PTEN, thus positively regulating PI3K/Akt [43,53,55,68]. Decreased cytoplasmic PTEN confers TMZ resistance by promoting proliferation in TMZ-resistant cells. How does Smurf1 promote TMZ resistance by synthetical regulation of nuclear and cytoplasmic PTEN? This study assumes that Smurf1 can ubiquitylate PTEN for both cytoplasmic degradation and nuclear import in response to TMZ. The phosphatase-inactivating mutation of PTEN, such as G129R and C124S, enhances the ability of PTEN nuclear import in response to IR [34]. Another study also confirmed that PTEN knockdown increases nuclear p-Akt [69], which may indicate that the reduction in nuclear PTEN increases DNA repair and leads to p-Akt nuclear transport. However, the interconnected determination of nuclear import by mono- or poly-ubiquitylation modification is still unclear. Using PTEN^K289E^ mutation to inhibit the mono-ubiquitylation of PTEN nuclear import, it is deduced that Smurf1 still degrades cytoplasmic PTEN^K289E^ protein levels to activate the PI3K/Akt signaling pathway under TMZ treatment. Smurf1-increased cytosolic PTEN leads to a decreased mTOR signal and further increases sensitivity to DNA damage. The present study suggests that Smurf1 interconnectedly regulates PTEN either to activate PI3K/Akt (degradation of cytoplasmic PTEN) or repair DNA damage (import of nuclear PTEN) to resist TMZ (Figure 4g). It is important to know whether targeting Smurf1 is effective in the TMZ resistance cell line. We also generated LN229-R cell lines and found that targeting Smurf1 could induce cell re-sensitivity to TMZ by protecting DNA damage and reducing the PI3K/Akt pathway in a PTEN-dependent manner. The MEFs from *Smurf1^+/+^* mice and *Smurf1^−/−^* mice, as well as the mice model of subcutaneous GB, also confirmed the combination of Smurf1 inhibition and TMZ treatment further inhibited tumor growth.

The combination of TMZ and Smurf1 inhibition may have a therapeutic effect on patients with PTEN wild-type GB. The dual PI3K/mTOR inhibitors, such as PI-103 and NVP-BEZ235, are synergistic with TMZ to increase the therapeutic efficiency [57]. In this study, the Smurf1 suppression can also inhibit the PI3K/Akt pathway. Moreover, phosphorylated Akt increases MGMT expression by reducing protein level and nuclear import transcription factor FKHRL1/FOXO3A [70]. Whether Smurf1 increases MGMT by activating Akt to resistant TMZ requires further investigation. Moreover, the miR-26a inhibitor synergistically decreases TMZ resistance by inhibiting the stemness of GSCs [71]. Smurf1 knockdown reduces the expression of stem cell markers (Sox2 and Nestin) by reactivating PTEN [43]. Whether Smurf1 inhibition regulates tumor stemness to re-sensitive TMZ requires further investigation. In addition, previous studies have shown that the PTEN deficiency synergistically works with PARP inhibitors by damaging DNA repair [72]. A further study warrants testing the combination of PARP inhibitor with TMZ in Smurf1 knockdown GB cells with the PTEN wild type. It is confirmed that the Smurf1 suppression not only inhibits the PI3K/Akt pathway but also increases the γH2A.X protein level in response to TMZ. Smurf1 is a novel target in TMZ resistance in PTEN wild-type cells, and it provides a new treatment scheme for TMZ re-sensitivity through multiple channels.

## 5. Conclusions

TMZ resistance is a major problem in the treatment of malignant brain tumors. However, the mechanisms underlying TMZ-resistant GB cells have not been fully characterized. Here, we report that TMZ-induced PTEN nuclear import depends on PTEN ubiquitylation by Smurf1. The Smurf1 suppression decreases the TMZ-induced PTEN nuclear translocation and enhances the DNA damage. In addition, Smurf1 degrades cytoplasmic PTEN^K289E^ (the nuclear-import-deficient PTEN mutant) in order to activate the PI3K/Akt pathway under TMZ treatment. Altogether, Smurf1 interconnectedly promotes both PTEN nuclear function (DNA repair) and cytoplasmic function (activation of PI3K/Akt pathway) to resist TMZ. These results provide a proof-of-concept demonstration for a potential strategy to treat PTEN wild-type GB patients who acquired TMZ resistance through targeting Smurf1.

## Figures and Tables

**Figure 1 cells-11-03302-f001:**
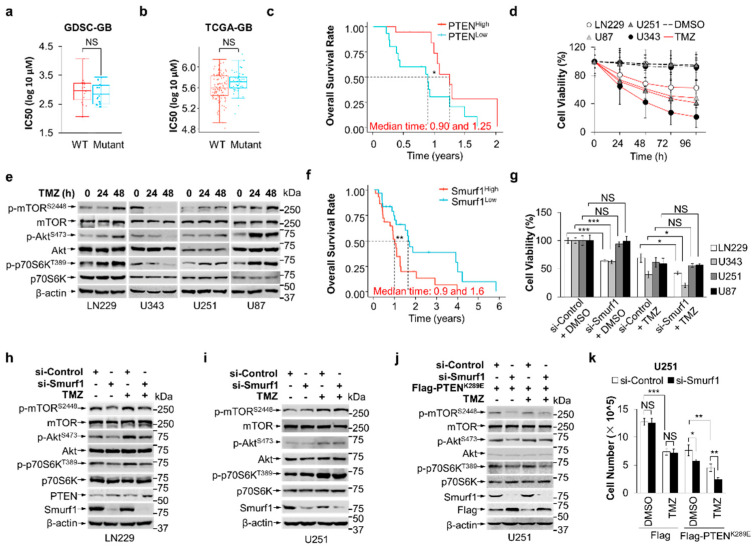
Combination of Smurf1 suppression with TMZ synthetically inhibits PI3K/Akt pathway. (**a**) The distribution of IC50 scores for TMZ in different-PTEN-status GB cell lines. The IC50 for TMZ and PTEN status of cell lines was obtained from the largest publicly available pharmacogenomics database (the Genomics of Drug Sensitivity in Cancer (GDSC), https://www.cancerrxgene.org/ (accessed on 19 September 2022)). PTEN wild-type group (wild type) included 8-MGBA, DBTRG05MG, SF268, GB-1, AM38, KS1, SNB75, YH13, LN18, LN229, DKMG, SW1088, U343, T98G, Becker, and SK-MG-1; PTEN mutant group (mutant) included U87, 42-MGBA, M059K, SF126, CAS1, U251, YKG1, U118, SF295, LN405, D263, LNZTA3WT4, D392-MG, D-542MG, D-247MG, D-566MG, and D-423MG. (**b**) The distribution of IC50 scores for TMZ in different-PTEN-status GB patients. RNA-sequencing expression (level 3) profiles and corresponding clinical information for GB cases were downloaded from the TCGA dataset (https://portal.gdc.com (accessed on 20 September 2022)). The GB cases were divided into PTEN wild-type group (wild type, *n* = 102) and PTEN mutant group (mutant, *n* = 47). Predicted the chemotherapeutic response for each sample based on GDSC. The prediction process was implemented by R package “pRRophetic”. The samples’ half–maximal inhibitory concentration (IC50) was estimated by ridge regression. All parameters were set as the default values. Using the batch effect of combat and tissue type of all tissues, the duplicate gene expression was summarized as mean value. The statistical difference between two groups was compared through the Wilcoxon test. (**c**) Kaplan-Meier survival analysis of the gene signature from TCGA dataset; comparison among different groups was conducted by log-rank test (* *p* = 0.034). The timeROC (v 0.4) analysis was used to compare the predictive accuracy of PTEN mRNA. Median time represents the time corresponding to the survival rate of 50% (i.e., the median survival time) in the high-expression group (1.25 years) and the low-expression group (0.9 years). (**d**) Cell viability was measured after the indicated time treatment with DMSO or TMZ (250 μM) in different GB cells (LN229, U87, U251, and U343) using MTT assay. (**e**) LN229, U343, U251, and U87 were cultured in the absence or presence of TMZ for the indicated time. The whole-cell lysates were examined through Western blotting for the expression of p-mTOR^S2448^/mTOR, p-Akt^S473^/Akt, p-p70S6K^T389^/p70S6K, and β-actin proteins. (**f**) Kaplan–Meier survival analysis of the gene signature from TCGA dataset; comparison among different groups was conducted by log-rank test (** *p* = 0.00902). The timeROC (v 0.4) analysis was used to compare the predictive accuracy of Smurf1 mRNA. Median time represents the time corresponding to the survival rate of 50% (i.e., the median survival time) in the high-expression group (0.9 years) and the low-expression group (1.6 years). All the analysis methods and R packages were implemented by R (foundation for statistical computing 2020) version 4.0.3. (**g**) MTT assay revealed the effect of Smurf1 knockdown on LN229, U343, U251, and U87 cells with DMSO or TMZ (250 μM, 48 h) treatment. (**h**,**i**) LN229 (**h**) and U251 (**i**) were transfected with si-Control or si-Smurf1 and then treated with DMSO or TMZ (250 μM, 48 h). Western blotting was employed to detect target proteins p-mTOR^S2448^/mTOR, p-Akt^S473^/Akt, p-p70S6K^T389^/p70S6K, PTEN, Smurf1, and β-actin. (**j**) U251 with Flag-PTEN^K289E^ was transfected with si-Control or si-Smurf1 and then treated with DMSO or TMZ (250 μM, 48 h). Western blotting was employed to detect target proteins p-mTOR^S2448^/mTOR, p-Akt^S473^/Akt, p-p70S6K^T389^/p70S6K, Flag, Smurf1, and β-actin proteins. (**k**) U251 with Flag or Flag-PTEN^K289E^ was transfected with si-Control or si-Smurf1 and then treated with DMSO or TMZ (250 μM, 48 h). The number of cells in each group were counted. In (**e**,**h**–**j**), Western blot analysis was performed in *n* = 3 biological replicates. In (**d**,**g**,**k**), data are presented as mean ± SD of three separate experiments, NS *p* > 0.05, * *p* < 0.05, ** *p* < 0.01, *** *p* < 0.001 as determined by unpaired two–tailed Student’s *t*-test.

**Figure 2 cells-11-03302-f002:**
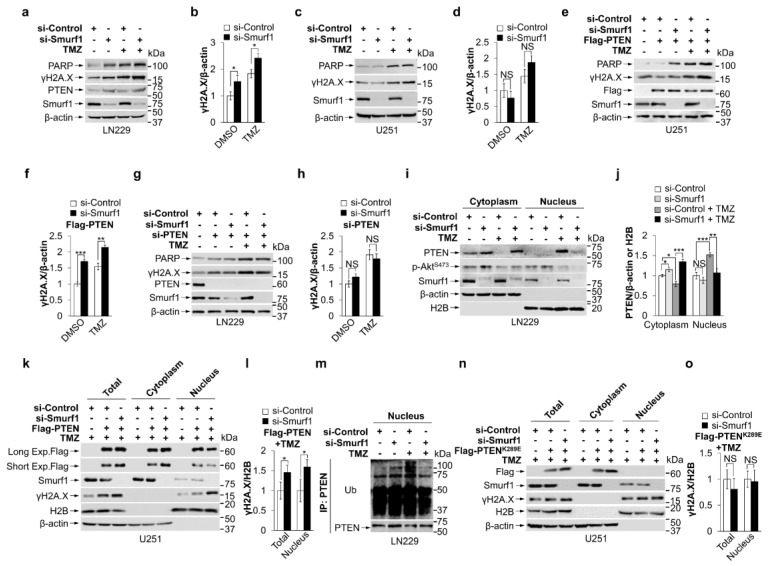
Smurf1 facilitates PTEN nuclear translocation to repair DNA damage under TMZ treatment (**a**–**h**): LN229 (**a**,**b**), U251 (**c**,**d**), U251 with Flag or Flag-PTEN overexpression (**e**,**f**), and LN229 treated by control or PTEN siRNA oligos (**g**,**h**) were transfected with si-Control or si-Smurf1 and then treated with DMSO or TMZ (250 μM, 48 h). Whole-cell lysates were examined by Western blotting with anti-PARP, anti-γH2A.X, and anti-PTEN antibodies (**a**,**c**,**e**,**g**). The overexpression or knockdown efficiency was examined with anti-Smurf1 or anti-Flag antibodies. β-actin was used as a loading control. The graph shows the relative γH2A.X intensity (**b**,**d**,**f**,**h**). Values were normalized against the number of cells treated with control siRNA oligos and DMSO. (**i**,**j**) LN229 cells were transfected with si-Control or si-Smurf1 and then treated with DMSO or TMZ (250 μM, 48 h). Cytosolic and nuclear fractions of cells were separated and subjected to Western blot analysis with anti-PTEN, anti-Smurf1, anti-p-Akt^S473^, anti-β-actin, and anti-H2B (**i**). The graph shows the relative PTEN intensity (**j**). Values were normalized against the amount of LN229 treated with control siRNA oligos and DMSO; (**k**,**l**) U251 with Flag or Flag-PTEN overexpression were transfected with si-Control or si-Smurf1 and then treated with DMSO or TMZ (250 μM, 48 h). Cytosolic and nuclear fractions of cells were separated and subjected to Western blot analysis with anti-Flag, anti-Smurf1, anti-β-actin, anti-H2B, or anti-γH2A.X (**k**). The graph shows the relative γH2A.X intensity (**l**). Values were normalized against the amount of Flag-PTEN U251 treated with control siRNA oligos and TMZ. (**m**) LN229 cells were transfected with si-Control or si-Smurf1 and then treated with DMSO or TMZ (250 μM, 48 h). The nuclear fractions were immunoprecipitated by anti-PTEN antibody. The immunoprecipitates were probed with anti-Ub and anti-PTEN antibodies. (**n**,**o**) U251 were transfected with si-Control or si-Smurf1, overexpressed Flag or Flag-PTEN^K289E^, and then treated with DMSO or TMZ (250 μM, 48 h). Cytosolic and nuclear fractions of cells were separated and subjected to Western blot analysis with anti-Flag, anti-Smurf1, anti-γH2A.X, anti-H2B, and anti-β-actin antibodies (**n**). The graph shows the relative γH2A.X intensity (**o**). Values were normalized against the amount of Flag-PTEN^K289E^ treated with control siRNA oligos and TMZ. In (**a**,**c**,**e**,**g**,**i**,**k**,**m**,**n**), Western blot analysis was performed in *n* = 3 biological replicates. In (**b**,**d**,**f**,**h**,**j**,**l**,**o**), data are presented as mean ± SD of three separate experiments, NS *p* > 0.05, * *p* < 0.05, ** *p* < 0.01, *** *p* < 0.001 as determined by unpaired two-tailed Student’s *t*-test.

**Figure 3 cells-11-03302-f003:**
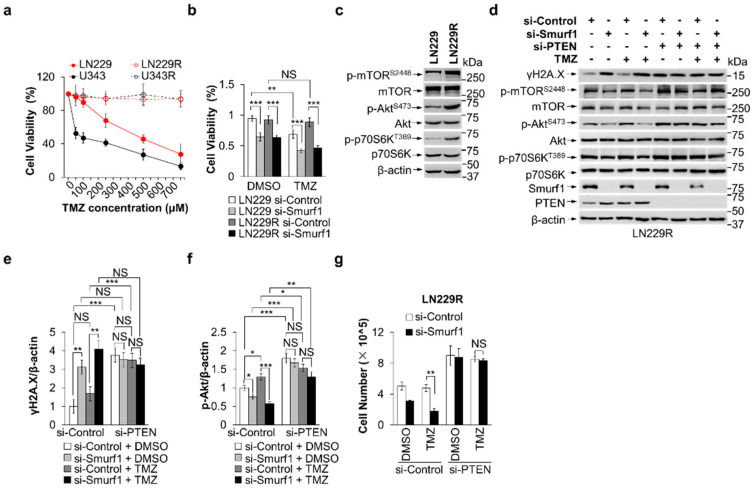
Smurf1 suppression restores the sensitivity of TMZ-resistant GB cells. (**a**) Cell viability was measured after 48 h treatment with increasing concentrations of TMZ in LN229, U343, LN229R, and U343R using MTT assay. (**b**) LN229 and LN229R were transfected with si-Control or si-Smurf1 followed by DMSO or TMZ (250 μM, 48 h) treatment. MTT assay was performed to evaluate the cytotoxicity of treatments on each group. Values were normalized against the amount of LN229 and LN229R treated with control siRNA oligos and DMSO. (**c**) LN229 and LN229R were prepared for whole-cell lysates and examined by immunoblotting with anti-p-mTOR^S2448^, anti-mTOR, anti-p-Akt^S473^, anti-Akt, anti-p-p70S6K^T389^, and anti-p70S6K antibodies. Blots were probed with anti-β-actin antibody to check protein loading. (**d**–**g**) LN229R cells were transfected with control, Smurf1, or PTEN siRNA oligos, followed by DMSO or TMZ (250 μM, 48 h) treatment. Western blotting was employed to detect target proteins γH2A.X, p-mTOR^S2448^, mTOR, p-Akt^S473^, Akt, p-p70S6K^T389^, p70S6K, Smurf1, PTEN, and β-actin (**d**). The graph shows the relative γH2A.X (**e**) and p-Akt (**f**) intensities. Values were normalized against the amount of LN229R treated with control siRNA oligos and DMSO. The number of cells in each group was counted (**g**). In (**c**,**d**), Western blot analysis was performed in *n* = 3 biological replicates. In (**a**,**b**,**e**,**f**,**g**), data are presented as mean ± SD of three separate experiments, NS *p* > 0.05, * *p* < 0.05, ** *p* < 0.01, *** *p* < 0.001 as determined by unpaired two-tailed Student’s *t*-test.

**Figure 4 cells-11-03302-f004:**
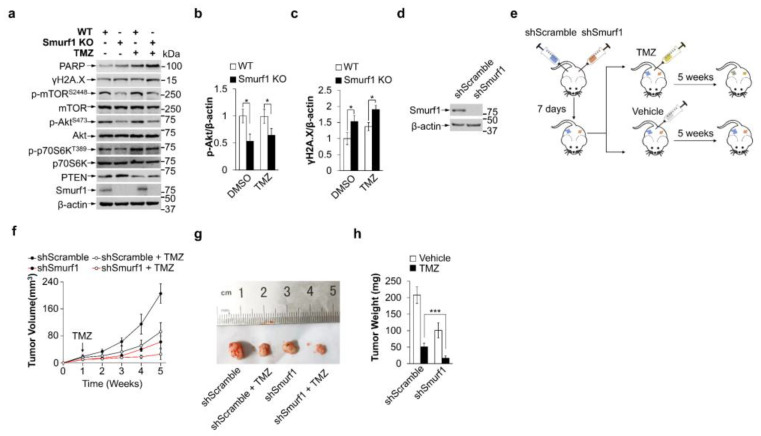
Smurf1 knockout restores sensitivity of PTEN wild-type GB cells to TMZ. (**a**–**c**) The MEFs were generated from *Smurf1^+/+^* mice and *Smurf1^−/−^* mice and treated with DMSO or TMZ (250 μM, 48 h). Cell lysates were prepared and subjected to Western blot analysis with anti-PARP, anti-γH2A.X, anti-p-mTOR^S2448^, anti-mTOR, anti-p-Akt^S473^, anti-Akt, anti-p-p70S6K^T389^, anti-p70S6K, anti-PTEN, anti-Smurf1, and anti-β-actin antibodies (**a**). The graph shows the relative p-Akt (**b**) and γH2A.X (**c**) intensity. (**d**) The knockout efficiency of LN229-shSmurf1 compared with LN229-shScramble cells was examined with anti-Smurf1 antibodies. Blots were also probed with anti-β-actin antibody to check protein loading. (**e**–**h**) Schematic showed timeline and procedure for the animal experiments. Male BALB/c nude mice were subcutaneously injected with LN229-shScramble or LN229-shSmurf1 cells (1 × 10^6^ cells per mice). Tumor-bearing mice were randomly separated into four groups when tumors reached 2–3 mm in diameter. Vehicle (75% ETOH, 25% PBS) or TMZ (20 mg/kg) was i.p. injected (every 2 days for 5 weeks). Mice were then euthanized and tumors (*n* = 3 per group in biological replicates) were removed (**e**). Tumor volumes were measured at different weeks (**f**). The representative picture was photographed (**g**), and their weights were measured (**h**). In (**a**,**d**), Western blot analysis was performed in *n* = 3 biological replicates. In (**b**,c,**f**,**h**), data are presented as mean ± SD of three separate experiments, NS *p* > 0.05, * *p* < 0.05, *** *p* < 0.001 as determined by unpaired two-tailed Student’s *t*-test.

**Figure 5 cells-11-03302-f005:**
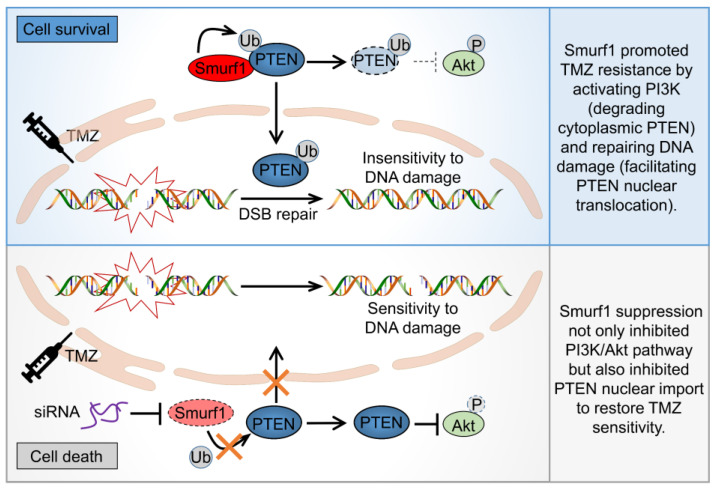
Model of synthetic effect of suppression of Smurf1 and TMZ treatment in PTEN wild-type GB cells. Model of GB restores sensitivity to TMZ by targeting Smurf1. Smurf1 interconnectedly regulates PTEN to activate PI3K/Akt (degradation of cytoplasmic PTEN) and repair DNA damage (import of nuclear PTEN), further promoting TMZ resistance. Smurf1 suppression not only inhibited PI3K/Akt pathway but also inhibited PTEN nuclear import to restore TMZ sensitivity.

## Data Availability

All data that can support the conclusions of this article are included in the article. The cell lines and plasmids generated in this study are available upon reasonable request. Further information and requests for resources and reagents should be directed to and fulfilled by the contact, Qin Xia (qin.xia@bit.edu.cn).

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
