# Peer review of "Smurf1 Suppression Enhances Temozolomide Chemosensitivity in Glioblastoma by Facilitating PTEN Nuclear Translocation"

_cells, 2022, doi:10.3390/cells11203302_

Round 1

Reviewer 1 Report (Previous Reviewer 1)

Dong et al. resubmitted their manuscript entitled “Smurf1 suppression enhances Temozolomide chemosensitivity by facilitating PTEN nuclear translocation”. Their major conclusions have not changed from the first submission of the manuscript. They have sent rebuttals to each of the reviewer critiques. Some of the rebuttals are well-received by the reviewer, but there still lacks overall scientific rigor that greatly diminishes the potential impact of the work. Furthermore, for some of the reviewer’s points, the authors unsatisfactorily addressed the reviewer concerns. Additional experimental studies need to be performed to conclusively tackle the questions raised by the reviewers. A point-by-point response to the rebuttal is addressed below.

A1a.    This question addressed whether the authors could produce a larger set of glioblastoma (GBM) cell lines to ascertain any baseline differences in sensitivity to temozolomide (TMZ) between PTEN wild-type and mutant cells. The authors do develop a panel of cell lines annotated for PTEN status. However, the level of detail provided to the reviewer is not sufficient and additional information is needed.

1)    Why is U323 not included in the list of PTEN wild-type cell lines and in the figure shown in the review document? This seems like an oversight, but since this is the most sensitive cell line to TMZ, this could alter the statistical conclusions that can be drawn.

2)    What are the units for the calculated IC50 in the Figure provided in the rebuttal? The reviewer cannot adequately address the rebuttal without knowing what the unit of measurement is.

3)    Furthermore, what assay was used to generate the IC50? Was it MTT? Cell counting/proliferation? It is unclear given the information given in the manuscript text and in the rebuttal what experiment was performed.

4)    What was the statistical test used for the rebuttal figure?

5)    The rebuttal figure should be included in the main text of the manuscript and in Figure 1. Furthermore, the information about cell line source and growing conditions for the additional cell lines used in the rebuttal figure should be included in the Materials and Methods.

There does appear to be a subset of cell lines that are sensitive to TMZ in both PTEN status groups. Is there anything about those cell lines that may be indicative of TMZ sensitivity? For example, the U343 cells were the most sensitive cell line to TMZ and had the highest expression of SMURF1 (in Appendix 1A). What does SMURF1 protein expression look like in the entire GBM cell line presented here?

A1b.    This question addressed the use of the LN229-TMZ resistant cell line. The authors did not adequately address the reviewers concerns about cell line usage. The rationale for using the LN229R cells remains underdeveloped. LN229R cells are already the most resistant cell line and so would not accurately model the development of TMZ resistance that would occur in a patient that had initial TMZ response. An additional acquired TMZ resistant cell line needs to be developed or obtained to verify these results in a cell line that was more sensitive to TMZ treatment at the beginning.

A2.      This response adequately addresses the reviewer’s concern.

A3.      This question addressed the incompleteness of control lanes in immunoblots. This concern is not sufficiently addressed. It is not sufficient to only include the proper controls in the Supplementary Information/Appendix for one set of experiments. The control lanes need to be paired experimentally. For example, in new Figure 2G, it is not clear what effect the siRNA targeting PTEN has on PTEN protein expression in LN229 cells. Also, in new Figures 2E, 2K, and 2N, the authors are assuming that the expression difference is the same compared to what is described in Appendix 2A. There needs to be clear controls for each experimental run that the authors are indeed increasing FLAG expression.

A4.      This question addressed the use of only one siRNA or shRNA targeting SMURF1 or PTEN. The response provided does not adequately address this concern. The usage of more than one siRNA and shRNA to target a gene is critical to sufficiently alleviate concerns about off-target effects. Even if you are getting your desired effect of depleting SMURF1 or PTEN, the siRNA or shRNA can also be causing other events within the cell (see Jackson and Linsley, Nat Rev Drug Discov, 2010). Using redundant siRNA or shRNAs will help alleviate these concerns. As such, the reviewer cannot discern fully whether the effects observed are on-target without any mitigating off-target effects.

A5.      This comment concerns the lack of rigor and transparency in the manuscript’s Materials and Methods section. Although the offers do provide some of the information requested, i.e. the source of U251 and U343 cells, the method detailing how the MEFs were cultured, site directed mutagenesis primer information, the MTT assay, and ubiquitin antibody details, there is still substantial data missing that was requested in the reviewer’s question as well as the need to clarify the rebuttal response.

1)    There is still no information given about cell line authentication and certifying the cells are mycoplasma negative. Using authentic human cell lines is critical for biomedical research and there are several options and services available that will authenticate cell lines based on STR profiles (see Almeida et al, PLOS Biol 2016). Cell lines infected with mycoplasma can give erroneous cell proliferation results, among other phenotypic alterations. Please acknowledge that the cell lines have been authenticated to be what they are claimed to be and that they were grown in a mycoplasma-free environment.

2)    Not all the cell lines listed in the reviewer figure that sought to answer question 1A are listed in the Materials and Methods. Please list the source and culturing conditions for those cell lines used in the Reviewer Figure. 

3)    For the shRNA experiments, was a non-targeting shRNA used or was it just the pLKO vector? A non-targeting shRNA (or an shRNA targeting a non-mammalian gene like the gene encoding luciferase) needs to be used instead of just the backbone vector.

A6.      This question concerned the quantification of gH2A.X immunoblots. The reviewer appreciates the author quantified the gH2A.X immunoblots depicted in the manuscript. However, that does not address the experimental request provided by the reviewer. The quantification of gH2A.X foci by immunofluorescence still needs to be performed as a complementary approach to justify the authors claims about the DNA damage response is altered by their studies.

A7.      Although the reviewer thinks the nuclear:cytoplasmic fractionation experiments for other members of the PI3K pathway would have greatly increased the impact of the manuscript, including this possibility in the Discussion is an acceptable compromise for this question.

A8.      This response adequately addresses the reviewer’s concern.

A9.      This comment addressed reviewer concerns about manuscript clarity, nomenclature usage, and terminology misuses. Unfortunately, there are still substantial areas of the manuscript where the English is not clearly written. This occurs frequently in areas with substantial editing between the first and second submission of the manuscript. For example, lines 27-8 (“more accessible to toxicity” can be “more toxic”) and line 31 (should include machinery or synonym after (MMR)) in the Introduction are not clear to the reviewer, and there appears to be some words missing in the transfection and MEF culturing information in the Methods. Furthermore, lines 204-6 and lines 215-9 among others in the Results section need additional editing. Please utilize professional English editing services for this manuscript.

A10.    Although the reviewer does not feel that this response adequately addresses the reviewer’s concerns about temozolomide sensitivity and proliferation rate, the reviewer concedes that this is beyond the scope of this work. The original Figure 1A from this first submission would be a better representation of the data. Upon review of the rebuttal, the rebuttal figure/new Figure 1A, and the manuscript text, it appears that the authors are just showing the response of the GBM cells to DMSO, which should be negligible.

A11.    This question addressed reviewer concerns about the baseline nuclear:cytoplasmic ratios of the cell lines used in this study. These concerns are only partially addressed. Specifically, the items below still need to be addressed:

1)    Given the data provided, it is not clear how the cells were binned into predominantly cytoplasmic, predominantly nuclear, or equally localized between cytoplasmic and nuclear cells. How was this quantified per cell? Did a cell have to be 100% nuclear or cytoplasmic PTEN to be in both? Did the cell have to be exactly 50% nuclear/50% cytoplasmic to be in the “equal” bin? More information is needed on the scoring criteria in the Methods. These results should also be confirmed with confocal microscopy as this will give a more precise and planar view of where PTEN is subcellularly localized more so than plain immunofluorescence microscopy.

2)    Also, this data suggests heterogeneity of PTEN localization within the cell line population. This should be commented in the Discussion as this is an interesting point.

3)    These data are also performed in cells already treated with control siRNA. Determining the baseline nuclear:cytoplasmic ratio in cells that are unperturbed by any foreign substance is needed.

4)    Furthermore, additional PTEN nuclear:cytoplasmic fractionation experiments need to be performed in the suite of PTEN wild-type cell lines now at the author’s disposal. As of right now, the authors are only describing a LN229 phenomenon.

5)    Please provide the PTEN antibody used to generate the rebuttal figure for immunofluorescence. Only the FLAG antibody is listed in the Materials and Methods.

A12.    This response adequately addresses the reviewer’s concern.

A13.    This response adequately addresses the reviewer’s concern.

Author Response

We are returning the revised manuscript entitled “Smurf1 suppression enhances Temozolomide chemosensitivity by facilitating PTEN nuclear translocation”.

These comments are valuable and have been very helpful for revising and improving our paper, as well as providing important guiding significance for our research. We have revised the manuscript according to the reviewers’ comments to make the paper more acceptable. This response letter explains point by point the details of our revisions in the manuscript and our responses to the reviewers’ comments. The main corrections in the paper and the responses to the reviewer’s comments are in the attachment.

Special thanks to you for your good comments.

We appreciate for Editors/Reviewers’ warm work earnestly and hope that the correction will meet with approval.

Once again, thank you very much for your comments and suggestions.

Qin Xia,

on behalf of the co-authors.

qin.xia@bit.edu.cn

Reviewer 2 Report (New Reviewer)

In this work, Dong and colleagues focused that TMZ-induced PTEN nuclear import depending on PTEN ubiquitylation modification by SMURF1. The SMURF1 suppression reduces the TMZ-induced PTEN nuclear translocation and improves DNA damage. In this way, Dong et al demonstrated that SMURF1 is associated with upgrades either PTEN nuclear function (DNA repair) and/or cytoplasmic function (activation of PI3K/AKT pathway) to resist TMZ.  In this way, Dong et al concluded that favorable outcomes can be achieved by the TMZ resistance in PTEN wild-type GB patients by targeting SMURF1.

I found this study interesting and relevant in this area.

The article is well written.

The working is fine and no further control is required.

I found the conclusion to be in line with the evidence and arguments presented.

The tables are fine.

The figures are fine.

I am only concerned about the caption of Figure 4 as it is a bit confusing. The authors should rephrase the caption of Figure 4. Also, concerning Figure 5, the authors must deliver Figure 5 in high quality.

Nice work!!

Author Response

Response to Reviewer 2 Comments

These comments are valuable and have been very helpful for revising and improving our paper, as well as providing important guiding significance for our research. We have revised the manuscript according to the reviewers’ comments to make the paper more acceptable. This response letter explains point by point the details of our revisions in the manuscript and our responses to the reviewers’ comments. The main corrections in the paper and the responses to the reviewer’s comments are as follows:

In this work, Dong and colleagues focused that TMZ-induced PTEN nuclear import depending on PTEN ubiquitylation modification by SMURF1. The SMURF1 suppression reduces the TMZ-induced PTEN nuclear translocation and improves DNA damage. In this way, Dong et al demonstrated that SMURF1 is associated with upgrades either PTEN nuclear function (DNA repair) and/or cytoplasmic function (activation of PI3K/AKT pathway) to resist TMZ.  In this way, Dong et al concluded that favorable outcomes can be achieved by the TMZ resistance in PTEN wild-type GB patients by targeting SMURF1.

I found this study interesting and relevant in this area.

The article is well written.

The working is fine and no further control is required.

I found the conclusion to be in line with the evidence and arguments presented.

The tables are fine.

The figures are fine.

Point 1: I am only concerned about the caption of Figure 4 as it is a bit confusing. The authors should rephrase the caption of Figure 4. Also, concerning Figure 5, the authors must deliver Figure 5 in high quality.

Response 1.1: Thank you for your suggestion, which plays a significant role in perfecting our paper. The caption of Figure 4 was changed to “Smurf1 knockout restores TMZ sensitivity of PTEN wild-type GB cells”. Figure 5 in high quality has been uploaded in the manuscript. All figures’ quality has been improved and confirmed.

Special thanks to you for your good comments.

We appreciate for Editors/Reviewers’ warm work earnestly and hope that the correction will meet with approval.

Once again, thank you very much for your comments and suggestions.

Qin Xia,

on behalf of the co-authors.

qin.xia@bit.edu.cn

Round 2

Reviewer 1 Report (Previous Reviewer 1)

I thank the authors for their point-by-point response to my review. I have included below a few additional questions/comments that need to be addressed.

1)    For the first point in the rebuttal, the comment that PTEN status made a difference in terms of temozolomide (TMZ) resistance in glioblastoma (GBM) patients does not seem completely justified by the results in Appendix Figure 1c. That data suggests that in wild-type PTEN GBM tumors, the levels of high SMURF1 gene expression correlate with decreased overall survival. This needs to be reflected with softer language in the main body of the text that clearly explain what is exactly shown in the Kaplan-Meier graph (lines 226-227). The authors also argue that the lack of differences found in PTEN wild-type and mutant tumors can be attributed to RNA sequencing only capturing the mRNA levels of a specific gene product. I wholeheartedly agree with this argument; however, I am not sure why this argument is made in both the main body of the manuscript and the rebuttal when the next data panel shown is TCGA RNA sequencing data. Perhaps mentioning in the Discussion or immediately after in the Results that understanding the protein levels of these proteins in patient tumors may further help stratify these patient populations. Furthermore, Appendix Figures 1a-c need to be brought forward to the main body of the manuscript as these data appear to be central to the overall narrative of the work.

2)    I appreciate that the authors added a second siRNA targeting SMURF1, but I would still like to see either a second siRNA targeting PTEN or a statement in the manuscript where siPTEN is used that “a caveat of the experiment is only one siRNA targeting PTEN was used limiting the conclusiveness of these results”, or something along those lines. The continued use of only one siRNA targeting PTEN still greatly reduces the impact and rigor of the study. These Figures include 2F-H, 3D, 3G, A1h-I, A2d, A2f, and A3f.

3)    Thank you for providing the STR profiles for U251 and LN-229. The authors state that the cells were tested for mycoplasma. Can the authors please confirm that the cells were mycoplasma negative?

4)    The gammaH2AX staining provided in Appendix Figure 2A/Rebuttal Figure 5 is intriguing. However, it is unclear based on the images provided whether the difference between TMZ treatment and TMZ+siSMURF1 truly is significant. This can only be achieved by quantifying the number of foci per cell. It is also unclear how many cells were imaged in the experiment. Furthermore, it is unclear what gammaH2AX antibody was used for the immunofluorescence experiment as that information is not included in the Materials and Methods section in the immunofluorescence sub-heading. Please provide that information.

All other review comments were adequately addressed.

Author Response

We are returning the revised manuscript entitled “Smurf1 suppression enhances Temozolomide chemosensitivity by facilitating PTEN nuclear translocation”.

These comments are valuable and have been very helpful for revising and improving our paper, as well as providing important guiding significance for our research. We have revised the manuscript according to the reviewers’ comments. The main corrections in the paper and the responses to the reviewer’s comments are the attachment.

Special thanks to you for your good comments.

We appreciate for Editors/Reviewers’ warm work earnestly and hope that the correction will meet with approval.

Once again, thank you very much for your comments and suggestions.

Qin Xia,

on behalf of the co-authors.

qin.xia@bit.edu.cn

This manuscript is a resubmission of an earlier submission. The following is a list of the peer review reports and author responses from that submission.

Round 1

Reviewer 1 Report

Herein, Dong et al set out to determine how regulating PTEN’s subcellular localization between the nucleus and cytoplasm in glioblastoma cells contributed to temozolomide chemosensitivity. Glioblastoma multiforme (GBM) is an aggressive cancer with few treatment options and dismal outcomes. Temozolomide (TMZ) is one of the standards of care agents to treat GBM, but TMZ resistance ultimately occurs either intrinsically or through acquired mechanisms. Understanding what contributes to TMZ chemosensitivity in GBM should positively and significantly impact clinical outcomes. PTEN is a tumor suppressor whose function is lost in approximately 35% of GBM tumors. PTEN exhibits both lipid and protein phosphatase activity and can be found in both nuclear and cytoplasmic compartments. PTEN localization dysregulation is an emerging mechanism that underlies how tumors can disturb PTEN’s tumor suppressive functions. Here, Dong et al suggest that the HECT-type E3 ubiquitin ligase SMURF1 (SMAD specific E3 ubiquitin protein ligase 1) regulates the TMZ-induced nuclear import of PTEN in GBM cells by ubiquitinating PTEN. Unfortunately, there are deficits in the experimental rigor that limit the impact of the presented findings, as well as sections of the text where the clarity in the wording further limits the potential impacts of the work. Cell line selection, failure to use contemporary or complementary techniques, and broad deficiencies in reporting reagent details are all areas that need to be addressed, among others. My major and minor comments are detailed below:

Major comments:

1)    The cell lines that were selected are not fully justified for the experiments that followed. As presented, there are four cell lines, LN229 and U343 are PTEN wild-type and U251 and U87 are PTEN mutants. The narrative of the study is mostly focused on using SMURF1 suppression as a method to overcome TMZ resistance in PTEN wild-type GBM cells. This claim is misleading based on the results compiled in Figure 1A. The two wild-type PTEN cell lines are at either extreme of sensitivity to TMZ as depicted in Figure 1A. The conclusions, then, are only describing a phenomenon seen in the LN229 cells and their TMZ-resistant derivatives. This is not sufficient to make broad, genotype-specific claims. Furthermore, it is unclear why making LN229-TMZ resistant cells, LN229R, would be beneficial for extrapolating acquired resistance mechanisms in Figure 3 when it is already the most TMZ-resistant cell line as shown in Figure 1A. There are two things that need to be done to address this point:

a.     A larger panel of cell lines with PTEN wild-type or mutant status annotated (perhaps in a Supplemental table) should be treated with TMZ to ascertain if there really is a relationship between PTEN status and TMZ response. At minimum, this will also determine a suite of cell lines to use as PTEN wild-type resistant cell lines.

b.     The authors should develop at least one additional TMZ-acquired resistant cell line from a PTEN wild type cell line that responds initially to TMZ treatment. The U343 cells would be a great cell line to use for this experiment.

2)    As a follow-up major concern to point #1 above, it is concerning to me that the responses shown in Figure 1A do not match up in other subpanels with similar experiments. For example, in Figure 1B there is no difference in sensitivity between the four cell lines shown to TMZ in the two siControl groups. This is also observed in Figure 3A. The LN229 sensitivity at 250 mM is different than what is shown in Figure 1A. Can this be explained by the authors?

3)    In immunoblotting subpanels Figures 1E, 1F, 1H, 1J, and 2D, there is incomplete information depicted. For example, there are no control lanes showing what protein expression for PARP, gH2A.X, FLAG-PTEN, SMURF1, and b-actin is for non-FLAG-PTEN expressing U251 cells. These types of controls are imperative to give the reader a sense of the magnitude of change seen in the cells caused by either the FLAG-PTEN overexpression (as is the case in 1E), or siPTEN knockdown in LN229 cells in 1F. As depicted, it is difficult to draw concise conclusions without the proper experimental controls. Along those same lines, in Figure 1F, there is no PTEN protein blot showing the extent of PTEN ablation following siRNA treatment. Therefore, no conclusions can be drawn as there is no way to examine how well the siRNA is depleting PTEN in the LN229 cells.

4)    Throughout the manuscript, only one siRNA or shRNA was used to target SMURF1 or PTEN. This is unacceptable and is not a contemporary experimental design. Multiple or pooled siRNAs or multiple shRNAs should be used to target SMURF1 or PTEN to help eliminate any possibility of off-target effects of the siRNA or shRNA. Using gRNAs to target SMURF1 or PTEN with CRISPR-Cas9 would also be an option. Using additional siRNAs/shRNAs/gRNAs for each experiment where those techniques were shown is needed to rigorously show that the authors have limited any potential claims being due to off-target effects.

5)    There is a significant lack of transparency or rigor in compiling the Materials and Methods. There is no mention of where the U251 or U343 glioblastoma cell lines were purchased or obtained. There is also no mention of how the Smurf1 WT and KO mouse embryonic fibroblasts were made or obtained or their culturing conditions. Furthermore, there is no indication of how any of the cell lines used in the manuscript were authenticated or tested for mycoplasma contamination. These are common practices for cell line studies in the cancer biology field. There is also no indication about how the site-directed mutagenesis for PTENK289E was done, such as what kit was used or what primers were used to mutate the DNA. The shRNA sequence is also not included for the shSMURF1 construct so it is unclear what part of the gene was targeted. There is also no information provided about how the MTT assay was performed. There is also no information given about the anti-ubiquitin antibody used in Figure 1I.

6)    For DNA damage experiments, gH2A.X is commonly visualized as foci that form at areas of DNA double stranded breaks. These foci are easily quantifiable by immunofluorescence and may be easier to discern changes in DNA damage. Please perform these gH2A.X foci immunofluorescence experiments as a complementary approach to illustrate how SMURF1 depletion in concert with temozolomide treatment or altering PTEN levels can affect the DNA damage response.

7)    There is emerging literature that there are AKT and mTOR signaling components found within the nucleus (Chen et al (2022) Nat Cell Bio or Dufour et al (2022) Cell Rep, for example). Please perform nuclear:cytoplasmic fractionation and/or immunofluroesence to show whether these pathways are localized to the nucleus or in the cytoplasm. This could have additional implications for the conclusions of this manuscript.

8)    SMURF1 is perhaps a genetic dependency in some human glioblastoma cell lines, including U343, according to the Cancer Dependency Map at the Broad Institute. The results in Figure 4 would be strengthened by including some of that in vitro data in that Figure as well as including more Discussion points about the possibility of SMURF1 as a glioblastoma target.

9)    The impact of the study is further limited by the lack of clarity in the writing and unacceptable uses of nomenclature, i.e. “ubiquitylation” instead of ubiquitination, non-capitalized protein names, not using italics for gene names. There would be great benefit to the reader if the authors sought assistance from an English scientific editor.

Minor comments:

1)    Since TMZ is a conventional chemotherapeutic agent, does TMZ sensitivity more closely correlate with baseline proliferation rate for these cells more so than PTEN mutation status? Please determine the proliferation rate, such as in population doubling experiments for the cell lines presented in the manuscript and correlate that doubling time with TMZ sensitivity and compare that to the correlation with PTEN mutation status.

2)    Baseline nuclear:cytoplasmic PTEN ratios need to be established for the cell lines used in the study. Only LN229 and U251 cells are used for these experiments and those experiments are done in the context of siRNA treatment. Simple fractionation or immunofluorescence experiments without any other perturbation should be performed to illustrate where PTEN is localized in GBM cell lines.

3)    Please add densitometry to the immunoblots. Many of the changes that are seen, although perhaps biologically significant, are hard to see. This is especially the case for the gH2A.X immunoblots in Figure 1C, 1D, 1E, and 1F. This quantification would make the readers job easier to discern the results the authors are trying to illustrate.

4)    The immunoblot in Figure 1I is too dark. A lighter exposure should be used.

Reviewer 2 Report

Dong et al. provide a biochemical overview describing the role of Smurf1 in regulating PTEN nuclear localization and it's effects in modulating Temozolomide sensitivity using glioblastoma cell lines and orthotopic mouse models. While the authors present some data that is potentially interesting, there are several aspects of the manuscript that require major revision:

- Extensive English language editing is needed. In the current form, there is awkward grammar and word use that makes the manuscript difficult to read and understand.

- The introduction and results are extremely difficult to understand. This may be in part due to English language deficiencies; however, the wording used to describe the major points is difficult to follow. There are also several instances of logical inconsistencies (for example: "enhanced nuclear PTEN promotes glioma sensitivity to chemo- or radiation therapy" (line 78-79) and "the inhibition of nuclear PTEN...enhances sensitivity to radiotherapy" (line 80-81) seem to directly contradict each other).

- The data is very complex with numerous mentions to several different cell lines, double treatment experiments, knockdowns, etc. When combined with verbose explanatory language and awkward English language use, the results become nearly impossible to interpret. The reader is left to either take the author at face value or spend inordinate amounts of time trying to decipher the text and figures. The manuscript would benefit from more clear, concise language and the figures can be organized better for increased clarity. For example, group experiments using the PTEN wildtype glioblastoma cell lines so that the reader doesn't have to constantly refer to which is which. Similarly, in Figure 4, the authors present a nice mechanistic diagram of their findings - this type of diagram would be helpful to introduce upfront so that the reader has a better sense for the interactions between the various proteins, how they are affected by TMZ, and what the downstream consequences are.

- There are numerous mentions of significant differences between western blot bands, however, there is no quantification presented to justify the actual significance of these differences. Many differences that were described as significant, at least to this reader's eye, seemed quite subtle. Additional quantification efforts would provide statistical weight to the authors' claims. 

-There are mentions to "Fig. S1a" (line 176) and "Fig. S1b and c" (line 186). Are these meant to refer to Figure 1 in the manuscript or is there a supplementary figure? If the former, then "Fig. S1b and c" does not refer to what the authors report (i.e. there are no PTEN knockdown results presented in Figure 1b or 1c).